# Role of Nasal Fibroblasts in Airway Remodeling of Chronic Rhinosinusitis: The Modulating Functions Reexamined

**DOI:** 10.3390/ijms24044017

**Published:** 2023-02-16

**Authors:** Jae Min Shin, Hyun Woo Yang, Jae Hyung Park, Tae Hoon Kim

**Affiliations:** 1Department of Otorhinolaryngology-Head and Neck Surgery, College of Medicine, Korea University, Seoul 02841, Republic of Korea; 2Upper Airway Chronic Inflammatory Diseases Laboratory, College of Medicine, Korea University, Seoul 02841, Republic of Korea; 3Mucosal Immunology Institute, College of Medicine, Korea University, Seoul 02841, Republic of Korea

**Keywords:** chronic rhinosinusitis, tissue remodeling, Fibroblast

## Abstract

Chronic rhinosinusitis (CRS) is a multifactorial inflammatory disease of the nose and sinuses that affects more than 10% of the adult population worldwide. Currently, CRS is classified into endotypes according to the inflammatory response (Th1, Th2, and Th17) or the distribution of immune cells in the mucosa (eosinophilic and non-eosinophilic). CRS induces mucosal tissue remodeling. Extracellular matrix (ECM) accumulation, fibrin deposition, edema, immune cell infiltration, and angiogenesis are observed in the stromal region. Conversely, epithelial-to-mesenchymal transition (EMT), goblet cell hyperplasia, and increased epithelial permeability, hyperplasia, and metaplasia are found in the epithelium. Fibroblasts synthesize collagen and ECM, which create a structural skeleton of tissue and play an important role in the wound-healing process. This review discusses recent knowledge regarding the modulation of tissue remodeling by nasal fibroblasts in CRS.

## 1. Introduction

Chronic rhinosinusitis (CRS) is defined as inflammation of the nasal passages and paranasal sinuses that lasts for more than 12 weeks. It is a significant health problem with a symptom-based prevalence of 5.5–28% in the general population [1,2]. Compared to acute rhinosinusitis, which is widely known to be a result of infection, CRS is an inflammatory disease caused by multifactorial conditions [3].

CRS is clinically subdivided into chronic rhinosinusitis with nasal polyps (CRSwNP) and chronic rhinosinusitis without nasal polyps (CRSsNP) according to the presence or absence of nasal polyps. In the last several decades, CRSsNP has been known to cause a neutrophil-dominant inflammatory response, whereas CRSwNP causes an eosinophil-dominant type 2 inflammatory response [4,5]. However, recent studies report differences in the mechanism of nasal polyps development between eastern and western countries, and many share similar pathophysiology among different phenotypes [6,7]. Therefore, endotypes subdivided based on pathophysiological mechanisms rather than phenotypes have received more attention.

CRS endotypes are classified according to the expression levels of inflammatory mediators involved in three representative immune responses (type 1, type 2, and type 3) occurring in the mucosal barrier. According to the latest evidence, it is suggested that different characteristics of upper respiratory tract remodeling occur depending on the endogenous type. Understanding this maybe offer great potential for customized treatment of CRS [8].

Tissue remodeling restores the structure and function of damaged areas in all organs of the body. It is a dynamic phenomenon that can cause temporary or permanent changes [9]. Ongoing inflammation due to dysregulated immune response can lead to pathological extracellular matrix (ECM) accumulation. This phenomenon was first reported in the lower respiratory tract, which is characterized by chronic inflammatory diseases such as asthma and chronic obstructive pulmonary disease [10,11].

Airway tissue remodeling is histologically characterized by the infiltration of inflammatory cells, overexpression of goblet cells, thickened basement membrane, and proliferation of fibroblasts, resulting in subepithelial ECM accumulation. Tissue remodeling also plays a key role in the pathophysiology of CRS, a representative chronic inflammatory disease of the upper respiratory tract, and is known to contribute to the recalcitrance of irreversible diseases [12].

Fibroblasts are spindle-shaped, diverse mesenchymal cells that play an important role in generating and maintaining the structural framework of the tissues. In particular, it has been found that during the healing process of tissue damage, it continuously secretes ECM precursors to aid in the regeneration and stability of connective tissue and to regulate the secretion of various inflammatory mediators [13]. Therefore, fibroblasts play a crucial role in production, remodeling, and contraction. However, few studies have clarified and reviewed the complex role of nasal fibroblasts in the pathological tissue-remodeling process in CRS. Thus, we would like to summarize the current knowledge on the process of tissue remodeling and signal regulation in CRS, with a particular focus on fibroblasts.

## 2. Normal Wound-Healing Process in Nasal Fibroblasts

The normal wound healing process in the human body proceeds in four steps: hemostasis, inflammation, proliferation, epithelialization, and remodeling [14]. Similarly, highly organized, well-regulated events occur during nasal mucosal injuries. First, to prevent bleeding caused in the early stages of tissue damage, the coagulation cascade is activated to create a blood clot and perform hemostasis. Subsequently, an inflammatory step occurs to prevent infection at the wound site. Fibroblasts, which quiescently reside in connective tissue before tissue damage, receive signals through chemoattractants secreted from the wound site and migrate to the wound area during the late inflammatory phase and early proliferative phase. For normal wound repair, migration and proliferation of nasal fibroblasts at this stage are essential for the next step [15]. Therefore, various studies have been conducted to identify factors affecting the migration of nasal fibroblasts and their related mechanisms. According to Hong et al., histamine stimulates the migration and proliferation of nasal fibroblasts, whereas antihistamine inhibits it [16]. Recent studies have shown that Prostaglandin E2 and steroids inhibit the migration of nasal fibroblasts [17]. Our study also demonstrated that smoking inhibits the migration and contraction of nasal fibroblasts, suggesting that this may interfere with the normal healing process after sinus surgery [18]. Taken together, the migration and proliferation of nasal fibroblasts are deeply involved in the wound-healing process of the nasal cavity and could be a therapeutic target for understanding the detailed mechanism.

After migrating to the damaged area, fibroblasts secrete matrix metalloproteinases (MMPs) to decompose fibrin clots generated during hemostasis and replace them with early granulation tissues such as collagen, glycoproteins, and proteoglycans [19]. During the inflammatory response, transforming growth factor-β (TGF-β) secreted by macrophages induces the differentiation of fibroblasts into active myofibroblasts. Myofibroblasts are characterized by actin filaments called α-smooth muscle actin (α-SMA) in the cytoplasm, and play an essential role in fibroblast migration, contractile forces, and ECM secretion. Activated fibroblasts secrete large amounts of enzymes, growth factors, and cytokines to rebuild the ECM microstructure and induce angiogenesis and epithelialization. During this process, ECM generation and remodeling occur via a dynamic balance between MMPs and tissue inhibitors of metalloproteinases (TIMPs) secreted by nasal fibroblasts [20].

The differentiation of nasal fibroblasts into myofibroblasts, ECM production, and tissue remodeling by TGF-β, a potent fibrosis-inducing cytokine, is well known [21]. However, Nonaka et al. argued that airway-remodeling patterns were altered by inducing heterogeneous responsiveness of TGF-β1 in nasal and lung fibroblasts [22]. In particular, it has been reported that TGF-β1 upregulates vascular endothelial growth factor (VEGF), which induces angiogenesis in nasal fibroblasts. Cho et al. suggested that VEGF was significantly increased in nasal polyp tissue and lipopolysaccharide (LPS)-induced nasal polyp-derived fibroblasts (NPDFs) [23]. Interestingly, Shimizu et al. reported that increased exosomes in fibroblasts co-cultured with eosinophils stimulated VEGF secretion [24]. In addition, the detailed mechanism and role of steroids, which are generally used as anti-inflammatory agents in suppressing the expression of VEGF in NPDFs and nasal fibroblasts, have been elucidated [25,26]. Therefore, regulation of VEGF expression may contribute to the wound-healing process in the nasal cavity and would provide evidence for the use of steroids in pathological tissue-remodeling diseases of the nasal cavity, such as CRS.

## 3. Remodeling Modulation of Nasal Fibroblasts in CRS

### 3.1. Nasal Fibroblast Activation

Pathogen-associated molecular patterns (PAMPs) of infectious factors such as bacteria, viruses, and microorganisms are recognized by pattern recognition receptors (PRRs) and toll-like receptors (TLRs) in cells, and activate innate immune responses [27]. In addition to infection, cell damage caused by physical injury and chemicals promotes the release of damage-associated molecular patterns (DAMPs) from cells, which bind to PRR, TLR, and IL-1 receptors to induce a non-infectious inflammatory response [28]. The innate immune response is primarily initiated when receptors such as PRRs and TLRs of immune cells such as monocytes, macrophages, neutrophils, mast cells, and natural killer cells combine with PAMPs and DAMPs to secrete primary inflammatory cytokines [28,29].

Secreted cytokines activate fibroblasts in the stroma of nasal mucosa. IL-1, IL-6, and TNF-α are pro-inflammatory cytokines that induce tissue inflammation by stimulating nasal fibroblasts [30]. IL-1β stimulates and activates nasal polyp-derived fibroblasts (NPDF) and increases CCL5 expression. CCL5 secreted from fibroblasts acts on blood vessels in tissues and induces trans-endothelial migration of immune cells, such as monocytes and eosinophils to the site of inflammation [31]. LPS from Gram-negative bacteria stimulates nasal fibroblasts, activates the TLR4/PI3K signaling pathway, and induces the expression of IL-6, IL-8, and MMP-1 [32]. Kang et al. reported that LPS induces the expression of TSLP through the TLR4/MAPK/NF-kB signaling pathway, which by corticosteroids and macrolide drugs [33]. Interferons (IFNs) are cytokines secreted by immune cells or fibroblasts as defense mechanisms when external viruses or pathogens invade the body [34]. Stimulation of fibroblasts by IFN-γ induces the secretion of RANTES, a chemoattractant factor for eosinophils, which means that fibroblasts promote eosinophil infiltration into tissues [35]. The epithelium is the first line of defense against external stimuli or antigens. The epithelial layer stimulated by PAMPs or DAMPs secretes epithelial-derived cytokines IL-25, IL-33, and TSLP, which causes fibroblast activation [36,37,38].

IL-6 regulates the immune response in the body. It is secreted by most cells, including fibroblasts, epithelial cells, endothelial cells, monocytes, T cells, and B cells [39,40]. IL-1, IL-6, TNF-α, IL-17, and TLR agonists amplify the NF-ĸB signaling pathway in fibroblasts and increase the expression of IL-6 [41]. Stimulated fibroblasts activate an autocrine loop that produces inflammatory mediators through leukemia inhibitory factor (LIF) and LIF receptor (LIFR), members of the IL-6 family [42]. Activated fibroblasts also increase the number of heterodimer receptors that accept IL-6 by upregulating the expression of the IL-6 receptor family and the gp130 core-receptor. Consequently, an autocrine effect on self-produced IL-6 becomes possible and more activated [42]. Activated fibroblasts have exhibit proliferation and excessive ECM accumulation. In addition to functioning as structural cells, they also regulate the immune response in local tissues through the secretion of proinflammatory cytokines, chemotactic factors, and various enzymes. Thus, it plays a role in regulating overall tissue remodeling, such as fibroblast stimulation of stroma, influx of immune cells, formation of vascular cells, and regulation of epithelial cells [41,43] (Figure 1).

### 3.2. ECM Accumulation

ECM is a three-dimensional network composed of collagen, laminin, fibronectin, elastin, and glycoproteins, including the stroma and basement membrane located beneath the epithelial layer [44]. CRSsNP, classified as non-eosinophilic or non-type 2, occurs predominantly in Asian regions such as Korea, Japan, and China. It demonstrates an excessive accumulation of ECM compared to normal tissues. Tissue remodeling is mainly caused and promoted by TGF-β1-induced myofibroblast differentiation in nasal fibroblasts and expression changes of the MMP/TIMP family [45].

Nasal fibroblasts are the most representative cell capable of producing ECM. They have a strong correlation with ECM overproduction. Under normal conditions, fibroblasts produce fibrillar components, fibronectin, and collagen types I, III, and V. TGF-β1 secreted from multiple lineages of leukocytes and stromal cells induces the differentiation of fibroblast into myofibroblasts that overexpress ECM [41]. We reported that TGF-β1-stimulated nasal fibroblasts overproduce ɑ-SMA, collagen types I, III, fibronectin, and total collagen, thereby inducing fibrotic properties of the stroma. It has also been reported that ECM deposition is characteristically observed in non-eosinophilic CRS. By contrast, no prominent ECM was observed in eosinophilic CRS or nasal polyps [21,46]. However, according to Wang et al., myofibroblasts were more abundant in the pedicle region of polyps than in the central or lateral regions of CRSwNP, which matched the TGF-β1 overexpression region [47]. In addition, Zhang et al. reported that many p-Smad2 positive fibroblasts were observed in the stalk of nasal polyps, indicating that the TGF-β1 signaling pathway in this region was amplified. In in vitro experiments, TGF-β1 increased α-SMA expression from 31% to 94.1%, and upregulated fibronectin. However, this mechanism was explained as a defense mechanism that prevents the spread of edema and inflammation and not as a trigger for nasal polyps [48]. Conversely, Radajewski et al. reported that the early development of nasal polyps was induced by the overproduction of ECM components, including periostin, by α-SMA-positive fibroblasts and myofibroblasts. IL-4 and IL-13, which are Th2 cytokines secreted from ILC2 and Th2 cells, stimulate myofibroblasts to promote the expression of ECM including periostin, proteoglycan, and collagen.

MMPs and TIMPs are closely related to the production and degradation of ECM proteins. An imbalance between MMPs and TIMPs induce excessive ECM accumulation and inflammation, contributing to extensive histological changes in CRS. Problems with non-destructive MMP function arise from excessive MMP expression [49,50,51]. 

TGF-β1 or TNF-α promotes the production of MMP-2 and MMP-9 in primary nasal fibroblasts through the NF-kB and smad2/3 signaling pathways. MMP-2 and MMP-9 degrade not only collagen types IV and V, but also elastin present in the ECM and basement membrane [52,53,54]. Schleimer et al. was reported that ECM components were upregulated in the CRSsNPs, and that the expression of these MMPs was driven by mesenchymal cells and fibroblasts [45]. Bachert reported that periostin induced the expression of MMP-3, MMP-7, MMP-8, and MMP-9 in NPDF, and high levels of MMP-9 were associated with a poorer prognosis after endoscopic sinus surgery [55]. However, in some studies, eosinophil accumulation and IL-4-, IL-5-, and IL-13-based inflammatory responses were significantly higher in MMP-9 knockout mice, and Th2 cytokines and eosinophils were decreased. These results suggest that MMP-8 and MMP-9 have anti-inflammatory functions and that MMPs increase as a defense mechanism to inhibit tissue remodeling [56]. TIMP is a major endogenous MMP inhibitor and relatively low expression of TIMP-1, and TIMP-2 has been observed in CRSwNP patients. In addition, studies have shown that MMP-2, MMP-3, MMP-7, and MMP-9 release TGF-β1 from the matrix, and the imbalance between these MMPs and TIMPs causes remodeling through ECM accumulation in tissues [57,58,59] (Figure 2).

### 3.3. Fibriolytic System Modulation; Fibrin Deposition

Blood coagulation, fibrin production, and dissolution are mechanisms that are known to occur naturally in the blood vessels in response to injury. However, the fibrinolytic system regulated locally in the upper airway induces tissue remodeling, such as fibrin deposition and edema [60]. When an inflammatory response is induced in tissues, VEGF is secreted from structural cells, such as epithelial cells and fibroblasts, inducing angiogenesis, and vascular permeability is increased by pro-inflammatory mediators [61,62]. This results in the tissue infiltration of plasma-clotting factors and eosinophils. In particular, excessive fibrin deposition occurs in eosinophilic CRSwNP because of an imbalance in fibrinolytic system-related proteins [63]. 

Thrombin activation downregulates the expression of tissue plasminogen activator (tPA) and upregulates factorXIII. This regulation inhibits fibrin degradation, resulting in excessive fibrin deposition in the tissues. Schleimer et al. confirmed that fibrin deposition was significantly increased in patients with CRSwNP rather than CRSsNP. The increased fibrin deposition was associated with reduced expression of tPA. In addition, when fibrinogen degradation production (FDP) was measured in the uncinate process of normal subjects and in the nasal polyp tissue of CRSwNP patients, it was found that the number of CRSwNP patients with CRSwNP was significantly reduced. This means that fibrin degradation was reduced, and excessive fibrin deposition occurred [64]. The enzyme tPA converts plasminogen into plasmin and degrades fibrin. It is mainly produced by epithelial cells, endothelial cells, macrophages, and fibroblasts. When tPA acts as a cytokine, it binds to the low-density-lipoprotein receptor-related protein-1 receptor (LRP-1R) in fibroblast and promotes the production of collagen and ECM. However, low tPA expression in nasal polyp does not mediate sufficient ECM production in fibroblasts, resulting in relatively low stromal density [65,66,67]. Conversely, fibrin deposition occurs because the plasminogen is not sufficiently activated. Such fibrin deposition is mainly observed in patients with eosinophilic CRSwNP, which is characterized by Th2-dominant inflammation. The concentration of t-PA has a negative correlation with ECP, an eosinophil-related protein. When bronchial epithelial cells and nasal epithelial cells were stimulated with the Th2 cytokines IL-4 and IL-13, the expression of tPA was significantly suppressed. However, in the case of uPA, there was no correlation with ECP concentration, suggesting that fibrin in the nasal mucosa is regulated by tPA rather than by uPA [5,64,68].

Plasminogen activator inhibitor-1 (PAI-1), an inhibitor of tPA and uPA, induces fibrin deposition in local tissues by inhibiting the fibrinolytic system [69,70]. Fibroblast is one of the cells that produces PAI-1, and it has been reported that TGF-β1 upregulates the expression of PAI-1 in fibroblast [71]. Takabayashi et al. reported that tPA expression was maintained at a fairly low level in nasal polyp tissues, whereas the expression of PAI-1 was not significantly increased. It was argued this i impact was insignificant. It has been reported that IL-4 and IL-13, which are Th2 cytokines secreted from ILC2 or Th2 cells, but not PAI-1, suppress tPA expression, and a correlation was confirmed in eosinophilic CRS patients [64] (Figure 3).

### 3.4. Epithelial Remodeling

The Wnt signaling pathway is essential for the differentiation of upper airway epithelial cells. However, upregulated Wnt signaling in CRSwNP leads to morphological changes or dysfunction of the epithelial cells [72]. Under normal conditions, β-catenin is degraded by GSK-3β. However, in the excessively activated Wnt pathway, β-catenin is not degraded and translocated into the nucleus to promote the cell cycle and proliferation [73]. In epithelial tissue, β-catenin exists in a form that binds to E-cadherin in epithelial cells. Upregulation of Wnt signaling induces EMT, which reduces E-cadherin expression in the epithelial layer and induces epithelial hyperplasia through β-catenin [8,74]. Dobzanski et al. found that the expression of WNT3A, a Wnt ligand, was increased in NPDFs, and when co-cultured NPDFs and epithelial cells, β-tubulin IV, a component of ciliary axons, and cilia were produced. It has been reported that the expression of FOXJ1, a transcription factor, is significantly decreased. This suggest that fibroblasts increase the epithelial permeability by reducing the cilia of the epithelium and elongating the cells [75].

The main sources of Th2 cytokines, including IL-4 and IL-13, are ILC2 and Th2 cells. However, several studies have reported that airway fibroblasts can be a source of Th2 cytokines. Nonaka et al. reported that nasal fibroblasts secrete IL-4 and IL-13 cytokines, and that fibroblasts have an IL-4 receptor and are stimulated by IL-4 to induce IL-6, TGF-β1, and C-C motif chemokine-11, which is reported to secrete monocyte chemotactic protein-4. IL-4 regulates the tight junctions of epithelial cells to increase their permeability and inhibit wound healing [76]. IL-13 induces goblet cell hyperplasia in epithelial cells and increases their proliferation of epithelial cells. Hyperplasia of goblet cells, regulated by IL-4 and IL-13, induces an imbalance between Muc5AC and Muc5B, disrupting the dynamic balance of mucin production [77,78]. In addition, elevated levels of Th2 cytokines in eosinophilic CRS induce mucin imbalance by activating the Notch signaling pathway in upper airway epithelial cells [79]. IL-6 is a representative cytokine produced by fibroblasts. IL-6 is overexpressed in CRSwNP and not only contributes to fibroblast activation of the local tissue immune response and regulation of stroma synthesis, but also regulates ciliary function by inducing epithelial proliferation and increasing cilia beating after epithelial damage [80].

EMT is a process by which epithelial cells transition into mesenchymal cells. Epithelial cells lose tight junctions, apical-basal polarization, and E-cadherin expression, but gain vimentin expression, motility, and the ability to produce the cytoskeleton and ECM [81]. EMT in the upper airway epithelial layer is regulated by transcription factors such as snails and slugs. Mechanisms that induce EMT in the upper airway epithelium have been reported to include the TGF-β1/Smad pathway, hypoxia, MAPK, AGE/RAGE, Wnt signaling pathway, and PPAR-γ [82,83]. IL-6 overexpression in cancer-associated fibroblasts (CAF) inhibits the phosphorylation of E-cadherin and β-catenin in epithelial cells and upregulates the expression of mesenchymal markers such as vimentin. This indicates that the production of IL-6 in stromal fibroblasts is associated with epithelial EMT. In addition, CAF promote the expression of growth factors such as EGF, TGF-β1, and FGF [84,85].

## 4. Cross-Talk between Nasal Fibroblasts and Immune Cells 

Fibroblasts are the key cells that regulate the activation or suppression of immune responses in chronic inflammatory environments [43]. In the past, it was recognized only as a traditional structural cell, but recently, it has been identified that the interaction between cytokines and chemokines secreted from fibroblasts and surrounding cells is a key factor regulating the immune response [86]. The epithelium is the first line of defence in the innate immune system. When foreign antigens or pathogens enter the epithelium, they recognize PAMPs through surface TLRs [87]. These PAMP and TLR interactions promote the expression of IL-25, IL-33, and TSLP in epithelial cells, which activates the immune response in tissues [88]. These epithelial cell-derived cytokines are associated with Th2 cytokine expression, mucus hypersecretion, and sub-epithelial fibrosis [89,90,91]. Park et al. reported that IL-25 induces the expression of MMP-1 and MMP-13 in fibroblasts in an IL-17RB-dependent manner. In addition, these processes were also related to the MAPK and NF-kB pathways and myofibroblast differentiation [36]. IL-33 can activate dendritic cells that induce naïve T cells into Th2 cells, and act directly on Th2 cells to induce the secretion of Th2 cytokines such as IL-5 and IL-13. When IL-33 is delivered to fibroblasts, it induces the expression of IL-6, which in turn induces the expression of ɑ-SMA and fibronectin in an ST2-dependent manner [37]. 

These mechanisms may be involved in tissue remodeling by inducing myofibroblast differentiation and ECM production. TSLP is master regulator of type 2 inflammation. TSLP stimulates dendritic cells to differentiate CD4+ T cells into Th2 cells. In addition, mast cells are stimulated with IL-1 to induce the secretion of th2 cytokines including IL-5 and IL-13 [92]. Although the mechanism by which TSLP regulates fibroblasts in the upper airway has not yet been clarified, TSLP in the lower airway upregulates the expression of collagen and ɑ-SMA in fibroblasts via the MAPK signaling pathway. Lee et al. identified the overexpression of TSLP via the MAPK and NF-kB pathways when nasal fibroblasts were stimulated with LPS [33].

Macrophages are involved in tissue remodeling and fibrosis and are a major source of MMPs and TIMPs. Macrophages stimulated by TLR ligands, IFN-γ, or GM-CSF mainly differentiate into M1 macrophages, whereas macrophages stimulated by IL-4 and IL-13 differentiate into M2 macrophages. M1 macrophages directly stimulate fibroblasts by stimulating the Th1 immune response and secreting pro-inflammatory cytokines such as TNF-α, IL-6, and IL-1β [93,94]. Stimulated fibroblasts are activated and maintained using autocrine and paracrine methods, and overproduce pro-inflammatory cytokines, such as IL-6, through the NK-kB pathway [43,95]. These fibroblasts induce tissue fibrosis by overexpressing the ECM along with differentiation into myofibroblasts or an increase in the number of cells. M2 macrophages are involved in wound healing, tissue remodeling, and inflammatory responses, and mainly produce cytokines such as IL-4, IL-10, and IL-13. M2 macrophages can also be activated by TGF-β1 and mediate the Th2 immune response [96,97]. In nasal polyps of CRSwNP patients, M2 macrophages were significantly increased, and there was a positive correlation with cytokines such as IL-5 and ECP. In addition, M2 macrophages secrete various growth factors such as TGF-β1, VEGF, and PDGF. The secreted growth factors contribute to the destruction of the ECM, which can worsen fibroblast proliferation and tissue remodeling [98]. M2 macrophages express FXIII-A, which regulates the fibrinolytic system. FXIII-A inhibits fibrin degradation, leading to excessive fibrin deposition in tissues [99].

IL-4, which is mainly secreted by ILC2 and Th2 cells, stimulates fibroblasts and induces the secretion of CCL17, which has a chemotactic effect on Th2 cells. In addition, it promotes eosinophil infiltration in tissues by inducing the expression of VCAM-1 in endothelial cells [100,101]. In experiments using nasal polyp-derived fibroblasts, fibroblasts stimulated with TLRs ligand secrete MCP-4, an eosinophil chemoattractant protein. This suggests that fibroblasts contribute to the inflammatory response in Th2 type or eosinophilic CRS [76,102,103]. Eosinophils are granular innate immune cells that play important roles in type 2 inflammation. Eosinophilia in the upper respiratory tract is associated with tissue remodeling and inflammation and induces EMT in bronchial cells to transform epithelial cells into mesenchymal cells. In our previous study, we investigated the effect of eosinophil-derived osteopontin on fibroblasts. Osteopontin stimulated nasal fibroblasts, increased the expression of pro-inflammatory mediators IL-6 and IL-8, and promoted ECM production. OPN expression was higher in uncinated process tissue than in the nasal polyp tissue of CRSwNP. Based on these results, osteopontin further accelerates inflammation in CRSwNP at early stage and acts as a trigger to induce tissue remodeling. On the other hand, Th2 cytokines such as IL-4, IL-5, and IL-13 secreted from eosinophils also regulate the fibrinolytic system through fibroblasts. It has been reported that the expression of tPA is kept low in CRSwNP patient tissue, and this is not the inhibitory effect of PAI-1, but the Th2 cytokine. Therefore, excessive eosinophil infiltration and secreted cy-tokines suppress tPA expression in structural cells, such as epithelial cells and endotheli-al cells, including fibroblasts. This action inhibits the activation of plasminogen and con-sequently blocks the mechanism of fibrin degradation, leading to the accumulation of fi-brin in the tissues [60,104].

Neutrophils are immune cells that are traditionally recruited first in an acute inflam- matory response and are known to be specialized in the elimination of invading pathogens. However, activated neutrophils contribute to the innate and adaptive immune responses by producing various cytokines [105]. The disruption and dysfunction of the epithelial layer that occurs in CRS makes it easier for the subepithelial tissue to expose antigens and induce an excessive immune response. Neutrophils form oncostatin M, contributing to the disruption of the epithelium of the nasal polyp, and stimulating fibroblasts in the stroma to induce IL-6 production and autocrine effects [80,106]. Epithelial cells exposed to various stimuli secrete CXCL8, which is an important cytokine that induces neutrophil infiltration. In addition, fibroblasts present in the nasal mucosa also secrete CXCL8 to recruit neutrophils into the tissue [5,107]. Fibroblasts stimulated with proinflammatory cytokines, such as IL-6, induce eosinophil infiltration by secreting chemotactic factor CCL2 (MCP-1) [108]. It has been reported that the IL-17 family, which is mainly secreted from Th17 cells, promotes the production of neutrophil chemotaxis factors such as CXCL1 and CXCL8, and the expression of IL-17A is increased in Asian patients with CRSsNP and CRSwNP [109]. Lee et al. reported that IFN-γ expression appears in neutrophils, which induces epithelial remodeling through p62-dependent apoptosis in nasal epithelial cells [110]. In addition, it secretes TGF-β1 to stimulate fibroblasts and induce myofibroblast differentiation, which causes stromal fibrosis. Differentiated fibroblasts also contribute to subepithelial fibrosis or EMT [111] (Figure 4).

## 5. Conclusions and Perspectives

Great progress has been made in our understanding of the function and role of fibroblasts over the past decades. Despite these advances, many questions remain, such as how nasal fibroblasts return to normal during the pathological remodeling process. However, recent studies have focused on epithelial cells and mucosal immunity, with relatively less interest in fibroblast-related research fields, tending to regard them as a legacy of the past.

Here, we summarize the detailed characteristics of tissue remodeling according to the endogenous form of CRS and the role of nasal fibroblasts in the process based on recent evidence. In summary, from the viewpoint of remodeling which induces intractability of CRS, fibroblasts play a key role in almost the entire process, and mediate and coordinate epithelial cells immune cells. Therefore, we need to reexamine the role of these nasal fibroblasts and continue research to improve our understanding of nasal fibroblasts based on new technologies and approaches. These advances will enable novel therapeutic approaches for the clinical prevention and tailored-treatment of pathological tissue remodeling in CRS.

## Figures and Tables

**Figure 1 ijms-24-04017-f001:**
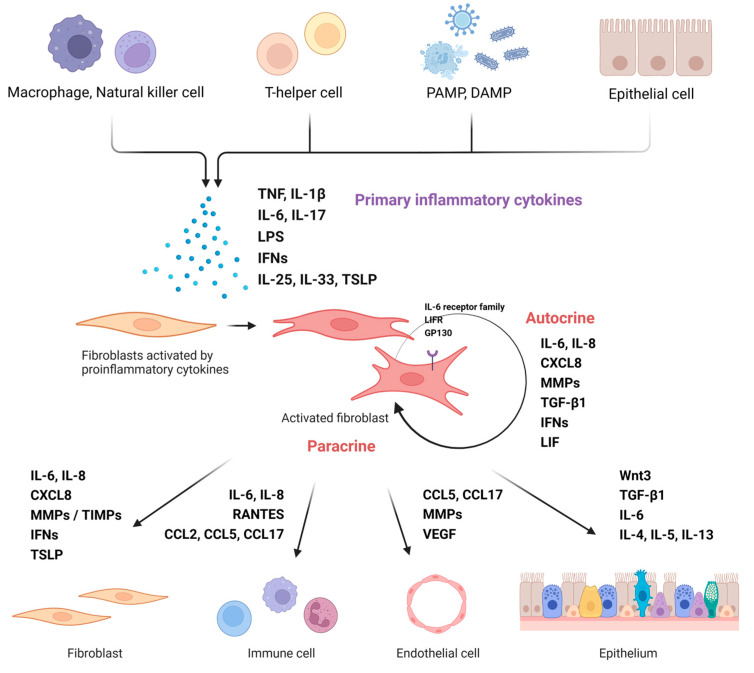
Nasal fibroblast activation and modulation of surrounding cells. Activation of nasal fibroblasts induced by immune cells or external antigens regulates various structural cells or immune cells in tissues. (Figure created using Biorender.com).

**Figure 2 ijms-24-04017-f002:**
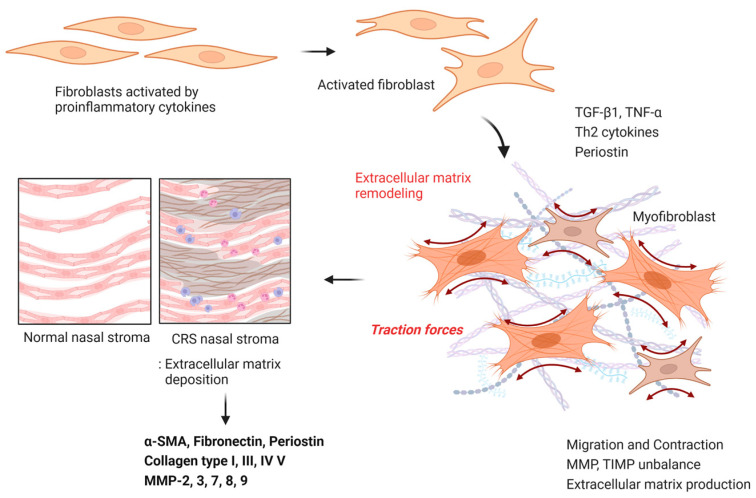
Myofibroblast differentiation and extracellular matrix deposition. Differentiated myofibroblasts have improved ECM generation, migration, and contraction abilities. The regulation of ECM, MMPs, and TIMPs produced by these cells affects tissue ECM accumulation.

**Figure 3 ijms-24-04017-f003:**
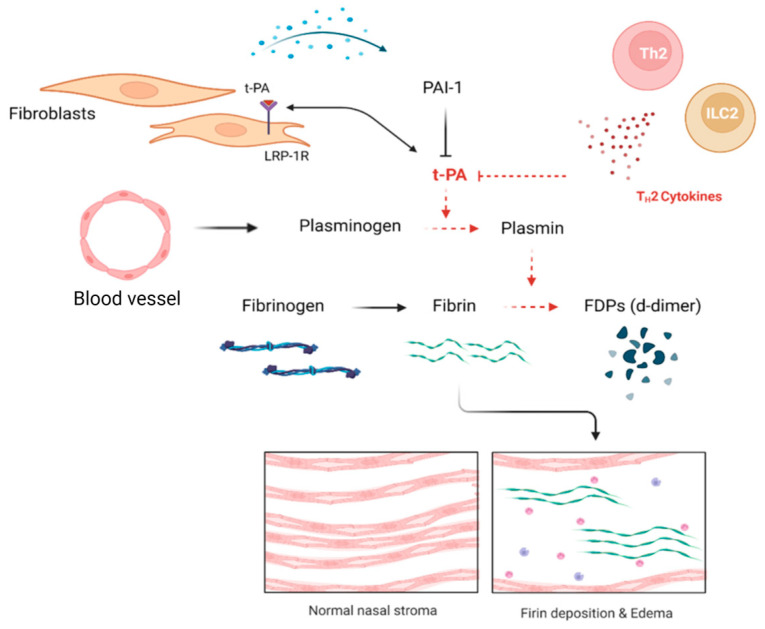
The role of nasal fibroblasts in the fibrinolytic system. The enzyme tPA binds to LRP-1R of fibroblast and induces the production of ECM such as collagen. However, fibroblasts activated by Th2 cytokine or other stimulators increase PAI-1 expression, and tPA suppressed by Th2 cytokine reduces ECM expression. In addition, a decrease in tPA causes an impairment of the fibrinolysis system and induces fibrin deposition.

**Figure 4 ijms-24-04017-f004:**
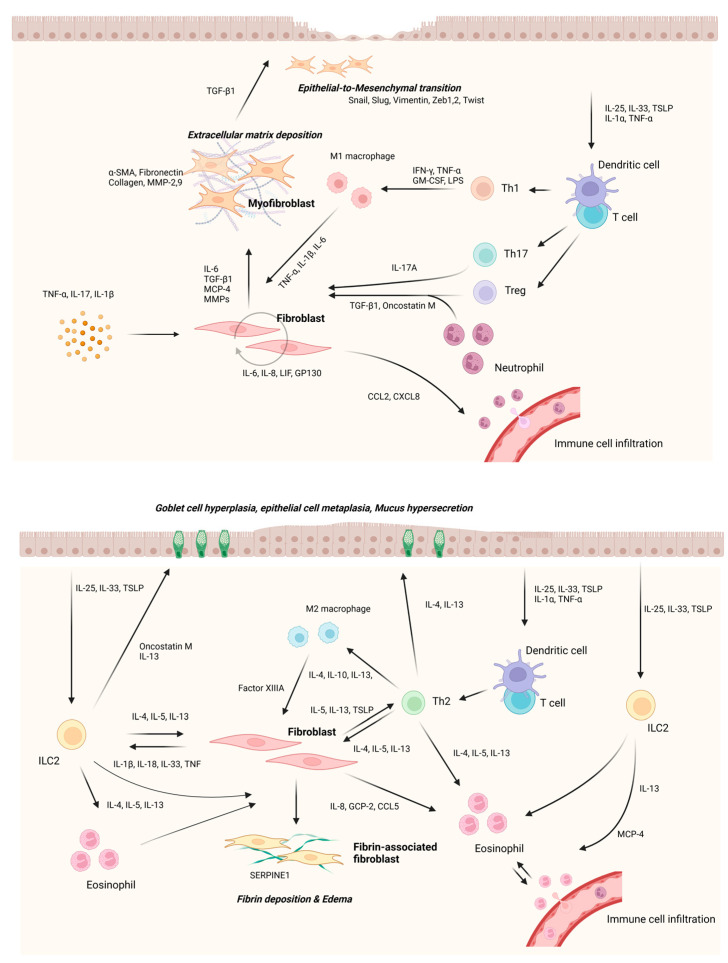
Interaction between fibroblasts and immune or other structural cells. Fibroblasts induce overall tissue remodeling of the nasal mucosa through cross-talk with neighboring cells. Interaction between nasal fibroblast and surrounding cells in Th1, Th17, or non-eosinophilic based immune response (upper panel). Interaction between nasal fibroblast and surrounding cells in Th2 or eosinophilic-based immune response (lower panel).

## Data Availability

Not applicable.

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
