# Peer review of "Role of Nasal Fibroblasts in Airway Remodeling of Chronic Rhinosinusitis: The Modulating Functions Reexamined"

_ijms, 2023, doi:10.3390/ijms24044017_

Round 1
Reviewer 1 Report
The article is a review about recent knowledge on the modulation of pathological tissue remodeling by nasal fibroblasts in Chronic Rhinosinusitis (CRS).
The process of Extracellular matrix (ECM) accumulation, Fibrolytic system modulation with Fribrin deposition, epitehelial remodeling and cross-talk between nasal fibroblasts and immune cells are very well explained with a decsription rich of many details.
Overall the article is very well written and what is decsribed is reported in the Figures which result highly explicative.
Author Response
To the Reviewer.
Thank you for taking your valuable time in reviewing the paper.
Some of the formats of our manuscripts were not suitable for IJMS, so we modified them to fit the format.

Reviewer 2 Report
Manuscript number: ijms-2188902-peer-review-v1 (1)
Title: Role of nasal fibroblasts in airway remodelling of chronic rhi-2 nosinusitis: The modulating functions re-examined 3
1. Yes, this subject is useful for publication in International Journal of Molecular Medicine
2. Author discussed recent knowledge regarding the modulation of tissue remodelling by nasal fibroblasts. The paper is review of literature, there is no own experience.
4. Discussion is logical and correct.
5. Conclusion is correct.
6. References are current and pertinent.
This paper should be published after revision:
- different type of chapter, compare 1. and 2
- there is no own experience.
Accept after minor revision. It depends on editor of Journal, i tis just only review.
Author Response
To the Reviewer
Thank you for taking your valuable time in reviewing the paper.
1) As you mentioned, some of our manuscripts were not suitable for IJMS. It has been modified according to the format. Thank you.
2) Reference papers 16, 18, 46, 52 and 107 are our research results.
We demonstrated the role of nasal fibroblasts in wound healing and the drugs or external environmental factors that inhibit it.
In addition, the effects of cytokines such as TGF-B1 on nasal fibroblast activation, myofibroblast differentiation, and extracellular matrix synthesis were investigated.
We also investigated the relationship between eosinophils and fibroblasts, and published a study that osteopontin secreted from eosinophils stimulates fibroblasts to induce tissue remodeling.
The results of these studies were mentioned in the review paper, and the review was written by interpreting them together with the results of other studies.
